# Biological Significance of the Protein Changes Occurring in the Cerebrospinal Fluid of Alzheimer’s Disease Patients: Getting Clues from Proteomic Studies

**DOI:** 10.3390/diagnostics11091655

**Published:** 2021-09-09

**Authors:** Cristina M. Pedrero-Prieto, Javier Frontiñán-Rubio, Francisco J. Alcaín, Mario Durán-Prado, Juan R. Peinado, Yoana Rabanal-Ruiz

**Affiliations:** 1Department of Medical Sciences, Ciudad Real Medical School, Oxidative Stress and Neurodegeneration Group, CRIB, University of Castilla-La Mancha (UCLM), Paseo de Moledores SN, 13071 Ciudad Real, Spain; CristinaM.Pedrero@uclm.es (C.M.P.-P.); Javier.Frontinan@uclm.es (J.F.-R.); franciscoj.alcain@uclm.es (F.J.A.); Mario.duran@uclm.es (M.D.-P.); 2Neuroplasticity and Neurodegeneration Laboratory, Ciudad Real Medical School, CRIB, University of Castilla-La Mancha (UCLM), 13005 Ciudad Real, Spain

**Keywords:** Alzheimer’s disease, proteomics, cerebrospinal fluid, CSF, biomarkers

## Abstract

The fact that cerebrospinal fluid (CSF) deeply irrigates the brain together with the relative simplicity of sample extraction from patients make this biological fluid the best target for biomarker discovery in neurodegenerative diseases. During the last decade, biomarker discovery has been especially fruitful for the identification new proteins that appear in the CSF of Alzheimer’s disease (AD) patients together with amyloid-β (Aβ42), total tau (T-tau), and phosphorylated tau (P-tau). Thus, several proteins have been already stablished as important biomarkers, due to an increase (i.e., CHI3L1) or a decrease (i.e., VGF) in AD patients’ CSF. Notwithstanding this, only a deep analysis of a database generated with all the changes observed in CSF across multiple proteomic studies, and especially those using state-of-the-art methodologies, may expose those components or metabolic pathways disrupted at different levels in AD. Deep comparative analysis of all the up- and down-regulated proteins across these studies revealed that 66% of the most consistent protein changes in CSF correspond to intracellular proteins. Interestingly, processes such as those associated to glucose metabolism or RXR signaling appeared inversely represented in CSF from AD patients in a significant manner. Herein, we discuss whether certain cellular processes constitute accurate indicators of AD progression by examining CSF. Furthermore, we uncover new CSF AD markers, such as ITAM, PTPRZ or CXL16, identified by this study.

## 1. Introduction

The cerebrospinal fluid (CSF) circulates around the brain and spinal cord, thus, constituting the most valuable biofluid for the identification of biomolecules associated with different brain pathologies, especially neurodegenerative diseases [1]. The number of studies aimed at understanding the composition in biomolecules with different origins in CSF and how they change during the development of brain associated diseases has increased strongly during the last decade [2,3]. Moreover, a basic search through the literature using MEDLINE (via PubMed) using the terms proteomics, CSF, and different brain pathologies reveals the significant increase that these studies have experienced during the last years (Figure 1), mostly due to the development of state-of-the-art technologies that have expanded the number of proteins and peptides identified in this biofluid from hundreds to thousands.

Specifically, in Alzheimer’s disease (AD), the most widespread neurodegenerative disease, more than 300 articles have addressed the search for CSF protein markers (Figure 1). The number of proteomic studies has increased during the last years and almost 100 of them have appeared since 2019. Several of these studies propose specific reliable biomarkers of AD in CSF [4] and several reviews have gathered information from them to demonstrate the consistent role for amyloid-β (Aβ1-42), total tau (T-tau), and phosphorylated tau (P-tau) [5,6], as well as other proteins such as VGF nerve growth factor inducible (decreased in AD) or CH3L1 (increased in AD) [7,8], and certain family of proteins, such as granins or pentraxins, as reliable hallmarks of AD relevant for diagnosis [6,9,10].

The recent exponential increase of new proposed candidates as protein biomarkers of AD in CSF evidence the necessity of a deeper analysis of the most consistent findings across studies. Therefore, we have compiled the most recent data from proteomic studies into a database in order to perform an analysis that takes into account the biological function of each protein, considering their role in the cell and their interrelationships. This study allowed us to identify underlying biological mechanisms in AD that appear reflected in CSF, as well as highlight emerging CSF biomarkers of AD.

## 2. Materials and Methods

The search for articles was limited between January 2012 and June 2021 (i.e., the last decade), and carried out using the PubMed and Google Scholar databases. The filter “humans” was applied and the following keywords were used for the search “cerebrospinal fluid” and/or “CSF”, “Alzheimer’s”, “biomarker”, “MS/MS”, “proteomics” and “mass spectrometry”. Notwithstanding this, our search retrieved several review articles and studies focused exclusively on unique proteins that were not included in the analysis.

A deeper analysis was performed in those articles to select AD CSF studies containing both quantitative information and a clear statistical analysis that corroborated their findings. Only those studies that compared healthy controls vs. AD were included, discarding the articles that compared samples of different diseases without including healthy individuals as controls. Following these criteria, a total of 30 articles were selected to generate a database (Appendix A). Appendix A lists the titles of each of the selected articles, the proteomic approaches used, and the number of proteins identified in each article with up- or down- expression in AD. To generate our database, proteins that showed consistent direction of change across different studies (proteins that appeared in at least two studies following the same direction) were considered for the analysis and classified according to their biological function (Table 1) and the information obtained on July 2021 for each protein in Uniprot and IPA (QIAGEN Inc., Hilden, Germany) databases. Network analysis and relevant metabolic pathways were explored using IPA, String (https://string-db.org, accessed on 5 July 2021) and Reactome (https://reactome.org, accessed on 5 July 2021). Reactome *p*-values are corrected for the multiple testing (Benjamini–Hochberg procedure) that arises from evaluating the submitted list of identifiers against every pathway [9]. For these analyses, it was considered whether the proteins were up- or down-regulated in the context of AD, which was crucial for the interpretation and organization of the information discussed in this review. Those proteins that showed the most consistent changes were explored in order to obtain accurate information regarding their biological function and its potential correlation with AD pathology. Especial attention was given to those proteins that constitute potentially novel biomarkers of AD.

## 3. Variability of the Proteomic Approaches

Several proteomic aproaches have been used during the last decade for the identification of biomarkers in AD CSF (indicated in Appendix A). As expected, the most used technique was LC-MS/MS, since nowadays is one of the most advanced approaches for biomarker identification [40]. Despite the existence of different proteomic approaches, several proteins were consistently found to be upregulated (Table 1). The better examples constitute CLUS (upregulated) and NPTXR (downregulated) which were found with similar trend in three different proteomic aproaches. Although we have to be cautious to compare different proteomic techniques due to the intrinsic differences in protein handling and identification, herein we verified that proteins such as TREM2, CSTN3, CD99, SODE, IBP6 and AMD display similar results irrespectively of whether LC-MS/MS or 2DE were employed (Table 1). Notwithstanding this, the specific information of proteins that were found to be altered in AD CSF as a consequence of the inclusion 2DE experiments is indicated in Table 1. Appendix A also shows now the predominant technique used in the selected articles.

## 4. Two Thirds of the of the Proteins That Change in the CSF of AD Are Intracellular

While the identification of biomarkers in CSF is crucial to the understanding of AD, unveiling the source of the change is essential to reveal the biological mechanisms that underlie this pathology. Contrary to what we initially expected, given the reduced number of cells present in CSF (approximately 5 cells per mL that mainly correspond to lymphocytes and monocytes); [41], our study reveals that 66% of the CSF altered proteins identified across studies are intracellular proteins, mostly cytoplasmatic (39%; Figure 2A). Amongst them, approximately 80% of the proteins (64) were increased in AD vs. controls. Taking into consideration the widely known neural death which occurs in AD [42], we would expect that a significant origin of these intracellular proteins in CSF may arise from remains of cellular debris that translocate to the CSF [43]. Furthermore, an increased infiltration of neutrophils has been observed during the development of AD. Indeed, increased expression of CD11b positive neutrophils, which are directly related to neutrophil migration, positively correlated with the severity of AD. In this line, a higher number of neutrophils were found in brain vessels of AD patients when compared with controls of similar age [44].

Platelets are considered biomarkers for early diagnosis of AD (as reviewed in [45]). Treatment of platelets with Aβ led to platelet activation and enhanced generation of reactive oxygen species (ROS) and membrane scrambling, suggesting enhanced platelet apoptosis [46]. Interestingly, among the top five pathways detected using Reactome, three of them were related to platelets. Specifically, this study has identified a total of 17 proteins directly related to platelet degranulation (*p*-value 2.1 × 10^−11^) and or platelet aggregation (*p*-value 1.1 × −10^−07^) (Appendix A). Whether this finding is related to increased apoptosis requires further investigation.

Together with the cells discussed above an additional source for membrane proteins are exosomes. In this sense, it was previously shown that 1 mL of human CSF contains ~2 μg of endogenous exosomes [47]. The finding that exosomes isolated from human CSF or brain samples sequestered oligomeric Aβ in the brain has led to propose their protective role in AD pathogenesis [47]. Indeed, several proteins of our database such as CHL1, KNG1, or APOA1 were found to be constituents of human CSF exosomes [48].

## 5. Increased Glucose/Pyruvate Metabolism in AD CSF

A variety of proteins related to glucose metabolism were found altered in CSF to the extent that glycolysis and gluconeogenesis constitute the two metabolic pathways most represented in our study (*p*-value 9.5 × 10^−11^ and 6 × 10^−9^, respectively; Figure 2B,C). Furthermore, all the proteins related to glucose metabolism appeared increased in AD CSF (z-score 2.65, Table 1). Most of the identified proteins are common enzymes for both processes, glycolysis and gluconeogenesis (Figure 3).

It is also important to mention the glycolytic enzyme pyruvate kinase M1/2 (KPYM), that catalyzes the synthesis of pyruvate from phosphoenolpyruvate [49] and the lactate dehydrogenase (LDH), which reduces pyruvate to lactate through a reversible reaction, thus allowing cells to generate or consume lactate depending on their metabolic profile.

Impaired glucose metabolism has been widely recognized as an early feature in the brain of subjects with AD since alteration of brain aerobic glycolysis is frequently observed in the course of AD [50,51]. It has also been proposed that reduced glucose availability in AD would force the brain to rely on gluconeogenesis (de novo synthesis of glucose). Interestingly, despite the low brain glucose uptake in AD, most post-mortem studies show consistent upregulation in glycolytic enzyme proteins [51,52], thus suggesting a compensatory mechanism for the low glucose supply in order to overcome a compromised mitochondrial function. In conclusion, evaluation of an increase of proteins directly related to glucose metabolism in CSF may reveal what takes place in surrounding tissues during AD progression.

## 6. RXR Signaling in CSF (LXR/RXR Activation Pathway)

One of the most relevant findings of this study is the significant reduction (*p*-value = 1.1 × 10^−10^) of proteins participating in the Liver X Receptor (LXR)/Retinoid X receptor (RXR) pathway in CSF from AD patients (z-score 2.7; Figure 2B,C). LXR/RXR activation pathway (Figure 4) is involved in a variety of processes associated with cholesterol metabolism, inflammation, oxidative stress, etc. [53,54]. Although LXR/RXR pathway represents a relevant pathway altered in AD [53,55], as far as we know, this is the first relevant mention of an overall reduction of LXR/RXR activation in AD CSF. In line with our observations, previous works have detected a reduction in the expression of LXR-β in plasma of AD patients compared to control samples [56]. Furthermore, the LXR/RXR pathway was also reduced in plasma samples of PSAPP and hTau mice models [57]. Retinoids modulate the expression of different key proteins in AD, as presenilin 1 (PS1), metalloprotease 10 (ADAM 10) or β-secretase [58,59]. Moreover, retinoic acid (RA) may have a central role in the pathophysiology of AD and reduced brain levels of this metabolite would constitute a risk factor for the development of the disease. Different mutations on the RA receptors can misregulate AD candidate genes such as PS1, ADAM 10, PS2 or APP [53].

As we pointed out, a reduction in the LRX activation was predicted with high confidence. This decrease might be related to the advanced development of the pathology, since LXR activation plays a key role in plaque reduction by increasing clearance, thereby improving cognitive impairment [60,61]. Furthermore, genetic loss of LXR in APP/PS1 mice induced a greater accumulation of Aβ plaques [62], while loss of LXR in healthy mice triggered neurodegeneration [63]. Therefore, decreased LXR/RXR pathway components in CSF may inversely correlate with Aβ deposition in tissues.

Another function of the LXR/RXR pathway comprises the activation of apolipoproteins that may serve as cholesterol acceptors [64]. Apolipoproteins constitute a family of proteins with a key role in transport and delivery of lipids, cholesterol homeostasis, and central nervous system (CNS) remodeling [65]. In this study, we observed decreased levels of different apolipoproteins (APOA1, APOC2, and APOL1) in CSF. APOA1, one of the most abundant proteins in human CSF, has been identified as part of senile plaques of AD patients’ brains [66]. A link between the presence of APOC2 and familial Alzheimer’s was observed decades ago [67], although, to date, its specific role in late-onset AD (LOAD) is still unknown. Similarly, APOL1 has been related to other pathologies such as kidney disease [68], while a direct link with AD has not been established yet.

## 7. Neuronal Function/Synaptogenesis

The organization of the proteins according to their function (Table 1 and Figure 2C) revealed that a high number of them were directly related to neuronal function and specially, synaptogenesis. Synaptogenesis is a dynamic process by which the formation and stabilization of synapses occur in the CNS [69]. In neurodegenerative diseases, such as AD, synaptic degeneration and synapse loss have been described as early events that precede neuronal death [70,71]. Indeed, several studies have proposed a role for synaptic proteins as specific biomarkers for AD in CSF [7,44,45].

The clearest reflection in CSF of the synaptic degeneration that occurs in AD is the family of ‘long’ neuronal pentraxins (NPTX1, NPTX2) and its receptor NPTXR since they constitute the most consistently decreased proteins in CSF and they display a direct function in neural differentiation [72], synaptogenesis [73] and synaptic plasticity [74,75,76]. Specifically, these proteins form mixed NPTX complexes which traffic to the extracellular surface at excitatory synapses where they interact with postsynaptic glutamate receptors [77,78]. Taken together, these three proteins were found to decrease in 8 different proteomic studies. The parallel and consistent decrease of these proteins in CSF would not fit with our previous suggestion of increased neuronal cell death. However, different studies have already described a down-regulation of NPTX2 in AD brains [79,80]. It has also been described that NPTX1 is accumulated in dystrophic neurites and surround plaques in postmortem AD brains [81,82,83], which would explain their decreased levels in CSF. Therefore, the inverse correlation of pentraxins amount in plaques/CSF could be a good indicator of neuronal death and/or synaptic loss.

Similarly, neurexins (NRX), one of the best-characterized families of presynaptic organizers, appeared reduced in 6 different studies. Likewise NPTX1, two of the three family members (NRX1 and NRX2) have been proposed as direct targets of Aβ oligomers [84]. In the same line, calsyntenin-3 (CSTN3), a direct NRX interactor [85,86] was also found to decrease in CSF. This transmembrane protein of the cadherin superfamily is distributed in postsynaptic membranes throughout the adult brain [87]. As occur with NPTX1, CSTN3 accumulates in dystrophic neurite surrounding Aβ plaques [88].

In view of these studies, we have to take into consideration that several proteins may not be released to the CSF and, oppositely, they might somehow be accumulated into the plaque in a process that could be carried out by Aβ and tau aggregation, thus ultimately driving a reduction of certain proteins in CSF. This may explain the variability observed across proteins not only directly involved in neuronal function but also those related to cell adhesion and components of the cell-matrix (Table 1 and Figure 2C). Although at different levels, 21 of these proteins decreased while 25 increased in CSF. The best example that reflects this heterogeneity raises from the members of the SPRC family. Based on our results, four members of this family are differently altered in AD CSF. This family of proteins comprises six members that present calcium-binding domains and regulate cell interaction with the microenvironment [89]. It is worth highlighting the potential role of SPRC on vascular pathology in AD. In the brain, SPRC is also expressed in endothelial cells, wherein it affects trans-endothelial permeability [89]. Moreover, since it acts as a chaperone of collagen IV through the SPRC-collagen binding domain, it has been observed a direct relationship between increased SPRC, collagen IV, and the thickening of the basal lamina of the cerebral vasculature, a feature commonly observed in AD brains [89,90,91].

Notwithstanding this, according to our study, SMOC1 was consistently upregulated in AD CSF in seven different studies (Table 1), thus making it a potential biomarker. Among its functions, SMOC1 promotes endothelial proliferation [92], and although it is overexpressed in AD brains, wherein it colocalizes with Aβ plaques [93,94], its specific role in AD is still unknown. Conversely, testican-1 (TICN1) was consistently downregulated in AD CSF. Different studies have shown a link between TICN1 overexpression in the brain and AD [95]. It surrounds Aβ plaques in brains of AD patients [95] and regulates proteins related to Aβ production and degradation, such as MMP2 or cathepsin-L [96,97]. Finally, it has been linked to APP sorting. Therefore, as we have previously proposed for other proteins, lower CSF levels of this protein may indicate an accumulation in the brain during the development of the pathology [95].

## 8. 14-3-3 Proteins Are up Regulated in CSF

The 14-3-3 family consists of seven highly homologous molecules that were first reported as regulators of tyrosine hydroxylase (TH) activity [98], four of which were consistently upregulated in the CSF of AD. These proteins have recently been linked to a variety of processes such as regulation of protein interaction and localization or transcription since they have a nuclear localization sequence [99]. It has been reported that 14-3-3 proteins regulate neuronal differentiation, morphogenesis and migration [100]. The expression of these proteins increases in cortical regions of AD patients and, even though they have not been observed in Aβ plaques, some evidence connects them with neurofibrillary tangles [99]. However, 14-3-3 family members cannot be considered suitable markers for differential diagnosis of AD [101,102] since they have also been detected in CSF of all dementia patients, thus suggesting their role as common markers for neurodegenerative diseases.

## 9. Cytokines and Hormones up and down Regulated in CSF

Five of the proteins in this section correspond to members of the granin family of proteins (chromogranins, secretogranins and VGF). These precursors of biologically active peptides and the products of their proteolitical cleavage have been proposed as biomarkers of different neurological diseases, included AD [103,104].

As expected, among them, VGF was the most consistent downregulated protein among the identified secreted proteins with biological activities. This protein is nowadays considered one of the best AD markers in CSF as it is consistently reduced in CSF of these patients [7].

Another remarkable protein that was consistently reduced in CSF was somatostatin (SMS). A long time ago it was proposed that diminished levels of somatostatin in CSF may be a specific AD signature compared to other neurodegenerative pathologies as Parkinson’s [105,106]. Whereas the reason for this reduction as a consequence of the pre-propeptide processing remains elusive, it has been recently reported that its deficiency has a direct link with a loss of integrity of the BBB in Aβ-induced toxicity [107].

On the contrary, among the four proteins in this group that appear increased, phosphoprotein 1 (SPP1, also called osteopontin) was found to be the most consistent upregulated protein (increased in seven different studies). This extracellular phosphoprotein is expressed in response to stress and injury and regulates macrophage infiltration and cytokine production [108,109]. In the past years, SPP1 has been linked to inflammation-associated neurological disease. Indeed, higher SPP1 levels have already been described in brain and CSF in AD patients [110,111].

## 10. Importance of Cofactors in CSF

Herein, we explored common affinities of the identified proteins for given cofactors. Interestingly, copper (Cu^2+^) emerged as the most common cofactor for several of the identified proteins (indicated in Table 1). Several meta-analysis have identified changes in Cu^2+^ concentration in brain and serum (reviewed by [112]), although conflicting evidence of copper’s role in AD has been pointed ([113,114], reviewed by [115]). It is well known that Cu^2+^ and Zn^2+^ interact with Aβ peptides with high affinity and these interactions have been proposed to accelerate Aβ_1-40_ and Aβ_1-42_ aggregation in vitro, thus contributing to their toxicity, ROS generation, and the development of Aβ neurotoxicity [116]. Nonetheless, the mechanism that Cu^2+^ utilizes to reach the brain is partially understood. Together with albumin (ALBU), the main protein needed for passive diffusion of Cu^2+^ through the blood-CSF barrier to the brain is ceruloplasmin (CERU) [117]. The fact that both proteins seem to behave inversely in CSF AD, showing decreased levels of ALBU and increased CERU, indicate that the equilibrium of Cu^2+^ transport in CSF is altered in AD.

Further evidence of the implication of Cu^2+^ in AD would be represented by metallothionein 3 (MT3), which regulates Cu^2+^ and Zn^2+^ transport and storage in CNS and inhibits their toxicity, thus representing one of the major players in metal homeostasis [118]. Conversely, the peptidylglycine α-amidating monooxygenase (AMD), a copper-dependent enzyme that regulates the secretory pathway, was found increased in CSF from AD patients. In mammals, AMD is essential to catalyze α-amidation, a necessary step to confer full biological activity to many neuropeptides [119,120]. Herein, we report for the first time the potential use of AMD as a consistent AD marker in CSF. Furthermore, to our knowledge, there is only one study investigating AMD in CSF where a reduction in enzyme activity in AD samples as compared to healthy, age-matched control was proposed, thus suggesting neuronal dysfunction within the CNS in AD patients [121].

## 11. Other amyloid β Interactors

The present analysis showed an increase in CLUS (APOJ) levels in CSF, which is in line with previous findings [122,123]. CLUS reduces aggregation and promotes clearance of Aβ at the blood-brain barrier under physiological conditions, suggesting that this protein, as proposed for SMS, may accumulate in CSF as a consequence of failed perivascular drainage of interstitial fluid [122].

Triggering receptor expressed on myeloid cells 2 (TREM2) has been shown to play a role in the phagocytosis of apoptotic neuronal cells [124] and it has been recently identified as a microglial Aβ receptor that transduces physiological and AD-related pathological effects associated with Aβ [125]. Several studies have shown that the loss-of-function mutations of TREM2 variants are linked to increased AD risk [126,127]. Additionally, increased TREM expression has been observed in AD patients suggesting an association between TREM2 levels and apoptosis in AD [128], though this is the first demonstration of its consistent increase in CSF.

A role for the metalloendopeptidase THOP1 as a potential β-secretase candidate was demonstrated by Koike et al., since it cleaves the full-length APP when overexpressed in COS cells [129]. This enzyme also promotes soluble Aβ degradation while no degradation was observed in aggregated Aβ [130]. THOP1 expression was significantly increased in human AD brain tissue as compared to non-demented controls, suggesting that increased THOP1 expression might be part of a compensatory defense mechanism of the brain against high Aβ load [131]. Herein, we have found a direct correlation of the data described in brain tissue with that of CSF.

## 12. New Potential AD Markers

Together with the above highlighted information for MT3, TREM2, THOP1 or AMD, other potential AD markers in CSF emerge from this study. Although several of our candidates have already been proposed as markers in post-mortem tissue, their relevance in CSF has not been unveiled. Therefore, the proteins mentioned below correspond to those unique proteins from Table 1 whose information regarding to AD, to our knowledge, is very limited or unexistent.

### 12.1. Ubiquitin C-Terminal Hydrolase L1 (UCHL1)

Only a few reports address the implications of UCH-L1 genetic variation in AD [132,133], and, although this is the first study that shows a recurrent increase of these proteins in CSF by proteomic approaches, its increase in CSF has been verified by other methodologies [134].

### 12.2. C-X-C Motif Chemokine Ligand 16 (CXL16)

CXL16 is produced by dendritic cells and it is known to attract lymphocyte subsets [135], especially, natural killer (NK) cells that express its receptor [136]. NK cells play an important role in the host defense, which is related to their ability to secrete a variety of cytokines and chemokines, as well as killing infected host cells (reviewed by [137]). Therefore, this marker might be a good target to detect AD-related neuroinflammation via CSF.

### 12.3. Protein Kinase C and Casein Kinase Substrate in Neurons 1 (PACN1)

PACN1 is required for the activity-dependent internalization of AMPA receptor, a key regulator of synaptic plasticity, which is thought to be one of the key cellular components underlying learning and memory. An increase in the number of synaptic AMPARs leads to long-term potentiation (LTP), whereas the removal of surface AMPARs by endocytosis results in long-term depression (LTD) [138]. PACN1 also regulates activity-dependent retrieval of synaptic vesicles in the presynaptic terminals [139]. It has been implicated in various neurodegenerative diseases (Parkinson’s disease and AD) [140,141].

### 12.4. Protein Tyrosine Phosphatase Receptor Type Z1 (PTPRZ)

A thorough search through the literature allowed us to identify one bioinformatic study that speculated the possibility that *PTPRZ* may regulate the cognitive and memory pathways through the CNS, thereby promoting the development of AD [142]. It remains to be investigated whether this observation is accurate, although our collected data clearly support this concept as we also found a concomitant increase of these phosphatases in AD CSF.

### 12.5. Integrin Subunit Alpha M (ITAM)

To our knowledge, this protein has never been associated with AD. Remarkably, a recent work studied genetic variability of 4 new genes, one of which encodes ITAM, which were predicted to contain variants associated with AD [143]. According to our analysis, this protein appears consistently increased across proteomic studies. Therefore, our study constitutes the first nexus between gene expression and protein increase of this integrin.

### 12.6. Myristoylated Alanine-Rich Protein Kinase C Substrate (MARCS)

MARCS is an interesting innovative marker of AD in CSF, since little is known about its relationship with AD. MARCS phosphorylation at Ser46 has been shown to constitute a hallmark of neurite degeneration [116] and there are studies that point towards the beneficial effects of increased MARCS levels for memory improvement [144].

## 13. Conclusions

Every year there is an increasing number of published articles that take advantage of evolving state-of-the-art technologies to identify/propose new biomarkers of neurodegenerative diseases. This is especially evident when we perform a literature search of proteomics, CSF, and AD. Therefore, a continuous revision of the results is mandatory to redefine with extreme accuracy new potential biomarkers that may help to identify prodromal or initial stages of the devastating neurodegenerative disease that represents AD. Herein, we aimed at understanding the importance of clusters of proteins that share similar functions in AD pathology also indicating a potential explanation of their up- or down-regulation in CSF. Although this is a study of a database generated with previous proteomic studies and bioinformatics analysis it must be taken with caution, we propose herein certain proteins as potential new biomarkers since they consistently appear up- or down-regulated in AD CSF. Whether proteins such as TREM2, THOP1, AMD, or ITAM constitute good biomarkers will be confirmed by further proteomics studies that will certainly appear.

## Figures and Tables

**Figure 1 diagnostics-11-01655-f001:**
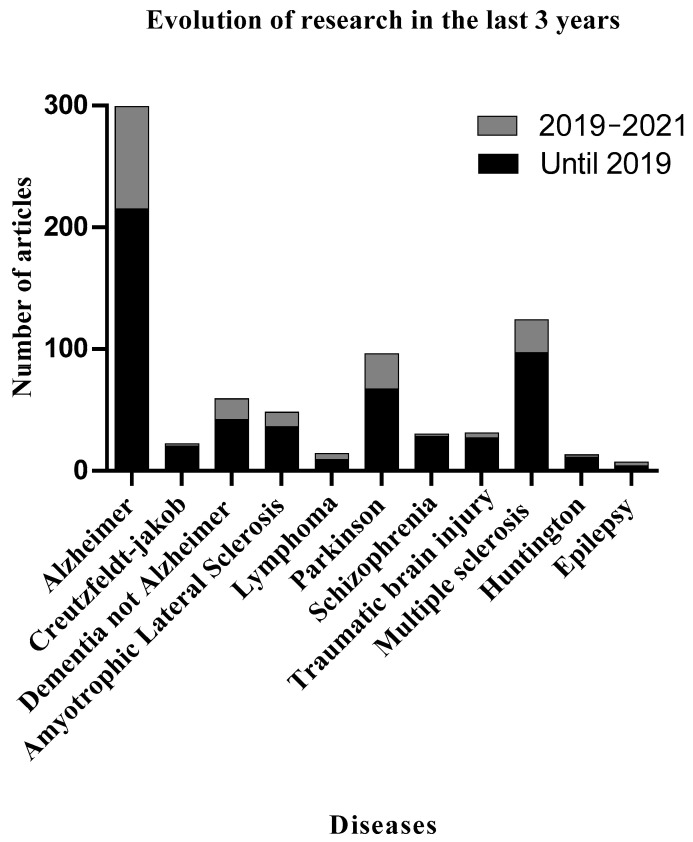
Articles published with the subject “proteomics, cerebrospinal fluid and different brain pathologies”. (Publications until June 2021). Articles published during the last three years are shown in grey.

**Figure 2 diagnostics-11-01655-f002:**
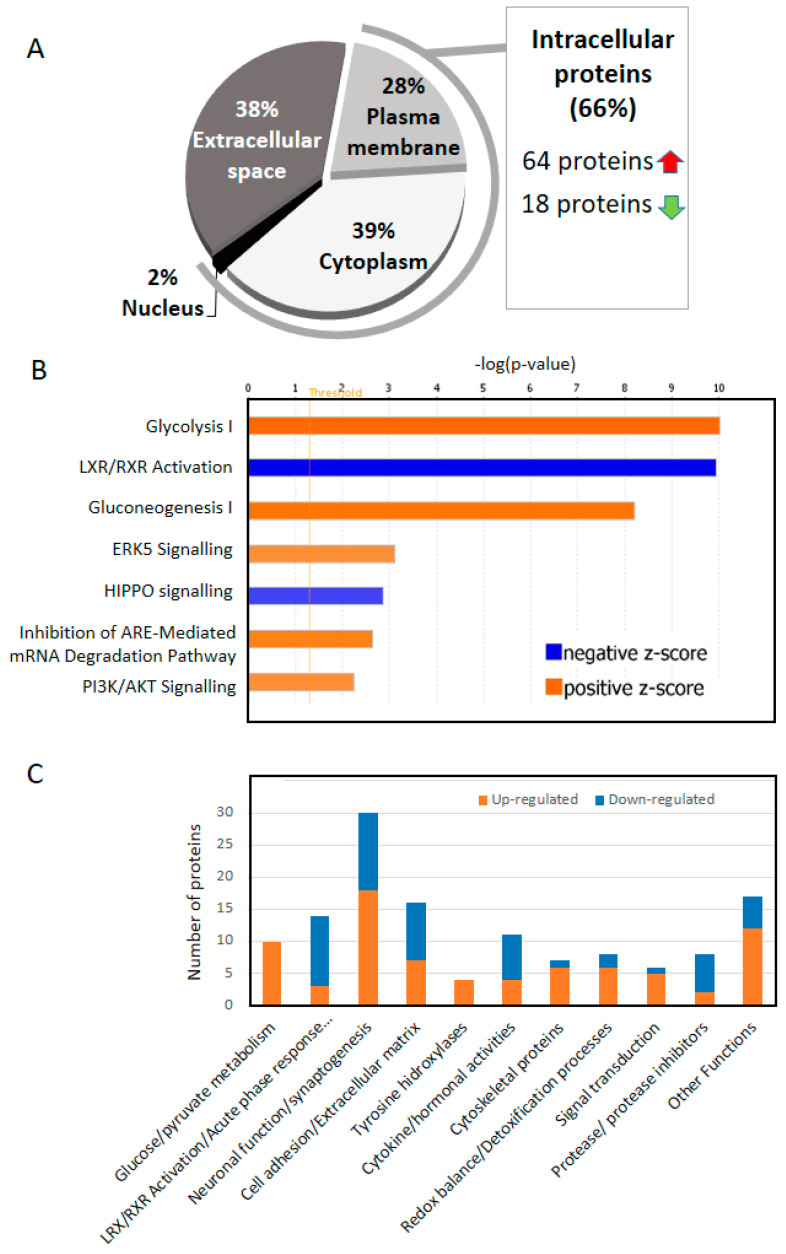
Information regarding proteins that consistently emerge across proteomic studies. (**A**) Schematic distribution of intracellular (plasma membrane and cytoplasm) and extracellular proteins (extracellular space). (**B**) Molecular pathways identified using IPA. Only those pathways with a –log(*p*-value) over 2 and a z-score of + or − 2 were considered. Positive z-scores are represented in orange. Negative z-scores are represented in blue. (**C**) Classification of the proteins according to their function. Data obtained from Table 1.

**Figure 3 diagnostics-11-01655-f003:**
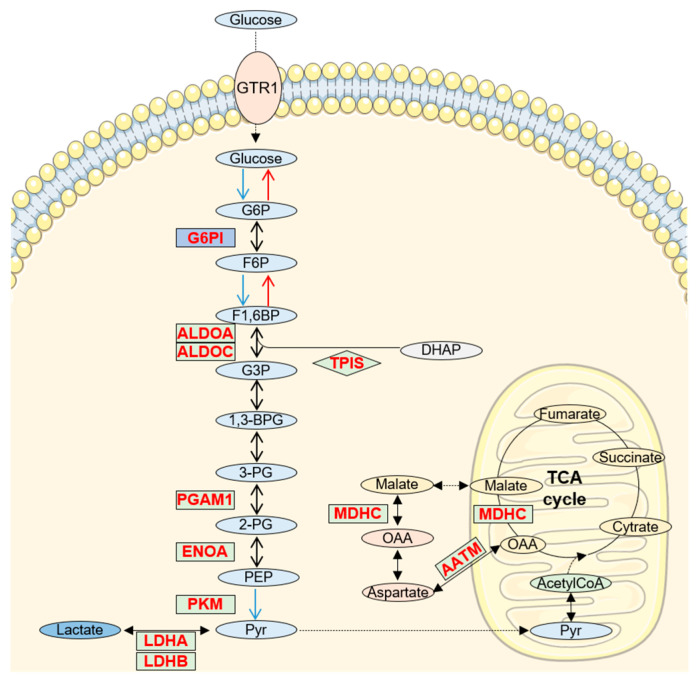
Alteration of glucose/pyruvate metabolism in AD CSF. Schematic representation of glycolysis, gluconeogenesis and pyruvate metabolism. Proteins that increase in AD CSF are labelled in red and green indicates decreased proteins. G6P: glucose-6-phosphate; F6P: Fructose-6-phosphate; G3P: glyceraldehide-3-phosphate; 1,3-BPG: 1,3-Biphosphoglycerate; 3PG: 3-phosphoglyucerate; 2PG: 2-phosphoglycerate; PEP: phosphoenolpyruvate; Pyr: pyruvate; OAA: oxalacetate. Background images were created using templates from Servier Medical Art, which are licensed under a Creative Commons Attribution 3.0 Unported License (http://smart.servier.com/ accessed on 5 July 2021).

**Figure 4 diagnostics-11-01655-f004:**
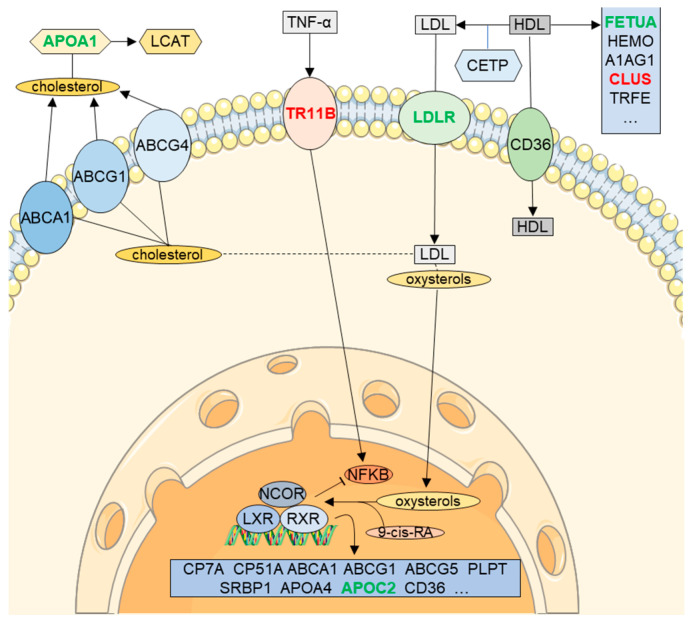
Alteration of LXR/RXR activation pathway in AD CSF. Summarized representation of the LXR/RXR activation pathway. Proteins that increase in AD CSF are labelled in red and green indicates decreased proteins. Background images were created using templates from Servier Medical Art, which are licensed under a Creative Commons Attribution 3.0 Unported License (http://smart.servier.com/ accessed on 5 July 2021).

**Table 1 diagnostics-11-01655-t001:** Proteins identified in CSF proteomic studies from Alzheimer’s patients that appear with similar direction of change in two or more articles. This analysis has been performed with the literature published during the last 10 years (see Materials and Methods).

Uniprot (Human)	Description	Identified in N Articles	References
GLUCOSE/PYRUVATE METABOLISM			
ALDOA	aldolase, fructose-bisphosphate A	6 (up) *	[10,11,12,13,14,15]
ALDOC	aldolase, fructose-bisphosphate C	2 (up)	[12,13]
ENOA	enolase 1	2 (up)	[12,13]
G6PI	glucose-6-phosphate isomerase	2 (up)	[12,13]
LDHA	lactate dehydrogenase A	3 (up)	[10,12,13]
LDHB	lactate dehydrogenase B	2 (up)	[16,17]
MDHC	malate dehydrogenase 1	5 (up) *	[10,12,14,18,19]
PGAM1	phosphoglycerate mutase 1	2 (up)	[10,12]
KPYM	pyruvate kinase M1/2	7 (up)	[11,12,13,14,15,16,20]
TPIS	triosephosphate isomerase 1	2 (up)	[12,17]
RXR SIGNALING (LXR/RXR ACTIVATION PATHWAY)			
FETUA	alpha 2-HS glycoprotein	2 (down)	[12,21]
ALBU ^#^	albumin	3 (down) *	[17,21,22]
AMBP	alpha-1-microglobulin/bikunin precursor	3 (down)	[12,17,23]
APOA1 ^$^	apolipoprotein A1	4 (down)	[17,22,23,24]
APOC2	apolipoprotein C2	2 (down)	[17,20]
APOL1	apolipoprotein L1	2 (down)	[11,12]
C1QB	complement C1q B chain	3 (down)	[10,17,25]
CLUS ^$^	clusterin	4 (up)	[19,22,26,27]
CERU ^#^	ceruloplasmin	4 (down) *	[11,12,13,17]
KNG1 ^Δ^	kininogen 1	3 (down)	[12,13,17]
LDLR	low density lipoprotein receptor	3 (down)	[12,20,28]
RET4	retinol binding protein 4	2 (down)	[12,17]
SODM ^Δ^	superoxide dismutase 2	2 (up)	[12,13]
TR11B	TNF receptor superfamily member 11b	2 (up)	[12,28]
NEURONAL FUNCTION/SYNAPTOGENESIS			
ACES	acetylcholinesterase (Cartwright blood group)	2 (up)	[10,12]
APLP1 ^$^	amyloid beta precursor like protein 1	3 (down) *	[17,21,26]
APLP2 ^$^	amyloid beta precursor like protein 2	3 (up)	[15,17,26]
A4 ^$^	amyloid beta precursor protein	3 (down) *	[17,26,29]
KCC2G	calcium/calmodulin dependent protein kinase II gamma	2 (up)	[12,13]
CSTN3 ^ε^	calsyntenin 3	3 (down) *	[15,17]
CPLX2	complexin 2	2 (up)	[10,12]
EPHA7	EPH receptor A7	2 (down)	[10,12]
NEUM	growth associated protein 43	4 (up)	[11,12,14,30]
GDIA	GDP dissociation inhibitor 1	2 (up)	[10,12]
MANF	mesencephalic astrocyte derived neurotrophic factor	2 (up)	[12,20]
MARCS	myristoylated alanine rich protein kinase C substrate	3 (up)	[10,12,14]
MT3 ^Δ^	metallothionein 3	2 (down)	[20,21]
NFL	neurofilament light	2 (up)	[12,20]
NFM	neurofilament medium	3 (up)	[11,12,26]
NPTX1	neuronal pentraxin 1	3 (down)	[25,31,32]
NPTX2	neuronal pentraxin 2	4 (down)	[12,20,25,33]
NPTXR	neuronal pentraxin receptor	5 (down)	[11,12,25,33,34]
NEUG	neurogranin	4 (up)	[12,20,30,33]
NRX1A	neurexin 1	4 (down) *	[12,21,25,31]
NRX2A	neurexin 2	3 (down)	[12,20,25]
NRX3B	neurexin 3	2 (down)	[20,31]
PACN1	protein kinase C and casein k. substrate in neurons 1	2 (up)	[11,12]
PCLO	piccolo presynaptic cytomatrix protein	2 (up)	[12,26]
RP3A	rabphilin 3A	2 (up)	[12,20]
RTN4	reticulon 4	2 (up)	[12,26]
SYUG	synuclein gamma	2 (up)	[10,12]
SYN1	synapsin I	2 (up)	[11,12]
TKNK	tachykinin precursor 3	2 (down)	[21,26]
TREM2 ^$,ε^	triggering receptor expressed on myeloid cells 2	2 (up)	[12,19]
CELL ADHESION/EXTRACELLULAR MATRIX			
C1QT1	C1q and TNF related 1	2 (down)	[11,12]
C1QT5	C1q and TNF related 5	2 (up)	[12,35]
CD99 ^ε^	CD99 molecule (Xg blood group)	2 (up)	[15,26]
NCHL1	cell adhesion molecule L1 like	2 (down)	[17,32]
FBLN3	EGF containing fibulin extracellular matrix protein 1	2 (down)	[17,20]
FBLN1	fibulin 1	3 (down)	[10,13,17]
ITAM	integrin subunit alpha M	3 (up)	[11,12,20]
MUC18	melanoma cell adhesion molecule	4 (down)	[15,17,25,32]
MMP2 ^Δ^	matrix metallopeptidase 2	2 (down)	[12,28]
NID2	nidogen 2	2 (down)	[12,13]
PGRP2	peptidoglycan recognition protein 2	2 (down)	[11,12,17]
SMOC1	SPARC related modular calcium binding 1	7 (up) *	[10,11,12,13,14,15,35]
SMOC2	SPARC related modular calcium binding 2	2 (up)	[12,28]
SPRC	secreted protein acidic and cysteine rich	3 (up)	[17,24,26]
TICN1	SPARC (osteonectin), cwcv and kazal like domains proteoglycan 1	3 (down) *	[10,17,21]
SPON1	spondin 1	3 (up)	[12,15,26]
14-3-3 PROTEINS			
1433B	tyrosine 3-monooxygenase/tryptophan 5-monooxygenase activation protein beta	3 (up)	[11,12,20]
1433E	tyrosine 3-monooxygenase/tryptophan 5-monooxygenase activation protein epsilon	5 (up)	[10,11,12,33,36]
1433G	tyrosine 3-monooxygenase/tryptophan 5-monooxygenase activation protein gamma	4 (up)	[10,12,13,20]
1433Z	tyrosine 3-monooxygenase/tryptophan 5-monooxygenase activation protein zeta	5 (up)	[10,11,12,13,33]
CYTOKINE (C)/HORMONAL (H) ACTIVITIES			
CCKN	cholecystokinin (H)	2 (up)	[13,26]
CMGA	chromogranin A (H)	4 (down) *	[21,31,34,36]
SCG1	chromogranin B (H)	3 (down) *	[12,17,21]
CXL16	C-X-C motif chemokine ligand 16 (C)	3 (up)	[10,20,26]
MIF	macrophage migration inhibitory factor (C)	2 (up)	[12,13]
SCG2	secretogranin II (H)	6 (down) *	[12,17,20,21,26,31]
SCG3	secretogranin III (H)	3 (down) *	[17,21,26]
OSTP	secreted phosphoprotein 1 (C)	7 (up) &	[11,12,13,15,18,19,26]
SMS	Somatostatin (H)	3 (down)	[12,20,26]
VGF	VGF nerve growth factor inducible (H)	11 (down) *	[11,12,21,24,26,31,34,36,37,38]
VIP	vasoactive intestinal peptide (H)	2 (down)	[12,20]
CYTOSKELETAL PROTEINS			
GELS ^$^	gelsolin	3 (down)	[11,12,20]
K22E	keratin 2	2 (up)	[11,12]
K1C9	keratin 9	2 (up)	[20,29]
MAP1B	microtubule associated protein 1B	2 (up)	[11,12]
MTAP2	microtubule associated protein 2	3 (up)	[11,12,20]
TAU	microtubule associated protein tau	4 (up)	[11,12,13,20]
STMN1	stathmin 1	2 (up)	[11,12]
REDOX BALANCE/DETOXIFICATION PROCESSES			
AATM	glutamic-oxaloacetic transaminase 2	2 (up)	[12,13]
GSHR	glutathione-disulfide reductase	2 (up)	[12,13]
GSTO1	glutathione S-transferase omega 1	2 (up)	[12,13]
PARK7	Parkinsonism associated deglycase	2 (up)	[12,13]
PPIA	peptidylprolyl isomerase A	2 (up)	[12,13]
PPIB	peptidylprolyl isomerase B	2 (down)	[10,17]
SODE ^#,ε^	superoxide dismutase 3	2 (down)	[10,15]
TRXR2	thioredoxin reductase 2	2 (up)	[12,20]
SIGNAL TRANSDUCTION			
IGF1R	insulin like growth factor 1 receptor	2 (up)	[12,20]
IBP6 ^ε^	insulin like growth factor binding protein 6	2 (down)	[12,15]
IMPA1	inositol monophosphatase 1	2 (up)	[12,13]
PEBP1	phosphatidylethanolamine binding protein 1	2 (up)	[10,12]
PTPRZ	protein tyrosine phosphatase receptor type Z1	3 (up)	[12,17,26]
SH3L3	SH3 domain binding glutamate rich protein like	2 (up)	[12,13]
PROTEASE/PROTEASE INHIBITORS			
CFAD	complement factor D	2 (down)	[17,20]
FETUB ^Δ^	fetuin B	2 (down)	[11,12]
PCS1N	proprotein convertase subtilisin/kexin type 1 inhibitor	5 (down) *	[10,17,21,25,32]
NEC2	proprotein convertase subtilisin/kexin type 2	2 (down)	[12,21]
ZPI	serpin family A member 10	2 (down)	[12,13]
THOP1	thimet oligopeptidase 1	2 (up)	[12,20]
UCHL1	ubiquitin C-terminal hydrolase L1	3 (up)	[11,12,19]
WFDC1	WAP four-disulfide core domain 1	2 (down)	[12,20]
OTHER FUNCTIONS			
ARP21	cAMP regulated phosphoprotein 21	2 (up)	[11,12]
VAS1	ATPase H+ transporting accessory protein 1	2 (down)	[12,17]
B3GN8	UDP-GlcNAc:betaGal beta-1,3-N-acetylglucosaminyltransferase 8	2 (down)	[11,12]
CH3L1	chitinase 3 like 1	7 (up)	[11,12,18,19,24,28,39]
CHIT1	chitotriosidase-1	3 (up)	[11,12,28]
TETN	C-type lectin domain family 3 member B	2 (down)	[12,29]
DDAH1 ^Δ^	dimethylarginine dimethylaminohydrolase 1	2 (up)	[12,20]
FABPH	fatty acid binding protein 3	4 (up)	[10,11,12,25]
FKB1A	FKBP prolyl isomerase 1A	2 (up)	[12,13]
GUAD	guanine deaminase	2 (up)	[11,12]
HBG2 ^Δ^	hemoglobin subunit gamma 2	2 (down)	[11,12]
HPRT	hypoxanthine phosphoribosyltransferase 1	3 (up)	[10,12,13]
OLR1	oxidized low density lipoprotein receptor 1	2 (up)	[12,20]
AMD *^#,^*^ε^	peptidylglycine alpha-amidating monooxygenase	2 (up)	[15,20]
PBIP1	PBX homeobox interacting protein 1	2 (down)	[17,20]
SLIT2	slit guidance ligand 2	2 (up)	[12,35]
SYWC	tryptophanyl-tRNA synthetase 1	2 (up)	[12,20]

The reference numbers correspond to those listed in Appendix A. *: Proteins identified with inverted pattern of expression in 1 additional article; &: Proteins identified with inverted pattern of expression in two additional articles; $: β-Amyloid-interacting proteins #: Proteins that bind copper; Δ: Proteins that bind metals. Down/up: protein down-up-regulated in the number of articles indicated; ε: Proteins that were identified at least in one article that used a less novel technique such as two-dimensional gel electrophoresis (2-DE); C: cytokine; H: hormone.

## Data Availability

Al data generated in this review are included in the article in the form of tables and Appendix A.

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
