# Peer review of "Biological Significance of the Protein Changes Occurring in the Cerebrospinal Fluid of Alzheimer’s Disease Patients: Getting Clues from Proteomic Studies"

_diagnostics, 2021, doi:10.3390/diagnostics11091655_

Round 1
Reviewer 1 Report
I am having a hard time comprehending this manuscript. I do not think that it offers anything new to the field, even as a review article. The authors need to do major revisions and include discussion on methodology, the instrumentation used to identify the proteins, and discussion on the quantification of the proteins before they can start deciphering the biological significance of these proteins. Discussion on the instrumentation is important because MS instruments, in particular, differ from each other and therefore there can be great changes in proteins identified.
Author Response
We appreciate the reviewer comments and the reviewer concerns regarding material and methods and discussion of the different instrumentation have been addressed point by point and are described below
- Material and methods:
We completely agree with the reviewer that the material and methods section should have been explained in more detail.
Indeed, this section has been rewritten and missing information, which is important to understand the discussion and the organization of the tables, has now been included. Specifically, we have inserted two paragraphs that contain a deeper explanation of the step-by-step procedure followed to come to our conclusions.
- Discussion on Instrumentation:
We have also included a whole paragraph as part of results/discussion (point 3) that deals with the heterogeneity of the instrumentation employed in each one of the articles included in this review. Furthermore, in order to make it easier to comprehend, the information corresponding to the proteomic technique used in each article has been included in Supplemental Table 1.
Specifically, proteins identified by less novel techniques, such as two-dimensional gel electrophoresis (2-DE), are now indicated in Table 1 with the symbol ε, which has also been included in the legend of the table.
Importantly, we have clarified in the material and methods section that, irrespectively of the technique that was employed, only those articles that contained a clear statistical analysis on their published results were included in this review.
Reviewer 2 Report
The authors in the manuscript entitled “Biological significance of the changes in CSF proteins of Alzheimer’s disease patients. Getting clues from proteomic studies” have explained about certain cellular processes constitute accurate indicators of AD progression by examining CSF. Also discussed about the new CSF AD markers, such as ITAM, PTPRZ or CXL16. The review is written concisely and briefly and provides significant detail on the topic which is scientifically sound. The manuscript has been written well and the content is comprehensive. Authors have discussed several points in the MS-like:
- They describe about the role of two thirds of the of the proteins that change in the CSF of AD are intracellular.
- How the Increased glucose/pyruvate metabolism in AD CSF provide the benefits of early diagnostics in AD.
- Interesting and valuable information on RXR signaling in CSF (LXR/RXR activation pathway).
- Neuronal function/Synaptogenesis and Tyrosine hidroxylases are upregulated in CSF discussed in this Review.
To conclude, this is a well-written and comprehensive review that makes a useful contribution to the field of AD research. The quality of the research is suitable for publication in the present form.
Author Response
We appreciate the positive comments of this reviewer. In this revised version we have included further details on material and methods and a new paragraph that deals with the heterogeneity of the proteomic approaches employed in all the articles included in this review.
Reviewer 3 Report
I was delighted to read this thorough narrative-style review of what amounts to be a straightforward or vote-counting-based research synthesis of CSF proteome studies in Alzheimer's disease. The review is well-written, and I particularly commend the structure of the materials by protein function, as well as a straightforward introduction of possible new candidate biomarkers.
My main concern, however, is with the way that the authors approached the literature search and associated reporting. While I realize that a formal meta-analysis was not the goal here and the effort itself undoubtedly deserves publication, I felt that the effort put into describing these Methods was insufficient to enable replication as well as to provide a proper contextualization for the findings.
Therefore, I highly recommend that the authors explicate their exact search terms and databases queried, as well as detail the inclusion and exclusion criteria and filters applied, as well as the selection of thresholds that capture 'Criteria for consistent change across different studies' (p.3). These studies likely need to be annotated with respect to the assaying technology used, extent of proteome coverage, and the implications (in the Limitations/Discussion) of these parameters for the presented findings and, in particular, enrichment and overrepresentation testing. Authors could also consider effect size pooling and a fixed effects meta-analysis as a straightforward extension of their effort, although I do feel this might be too demanding of a task.
I highly recommend the authors consult with and follow guidelines developed specifically for this purpose, such as PRISMA (http://www.prisma-statement.org).
A minor point: authors correctly mention that CSF is a treasure chest for biomarker discovery in neurodegenerative disease. One interesting implication is whether the invasiveness of this testing can be reduced by considering proteins whose plasma abundance is in high concordance with CSF levels and for which sensitive assays have been developed recently to adequately quantify their presence (e.g., proliferating NFL assays) even at low abundance levels.
Overall, I feel that the study is of clear value to the field, and the manuscript is an easy and informative read. My issues with the study primarily concern the review methodology and associated underreporting, but these can be ameliorated easily. I will be happy to review the revision of this manuscript.
Author Response
We are happy that the reviewer really enjoyed the reading of the review, and we appreciate the kind words.
The main concern of this reviewer is related to the reduced information included in the material and methods section. We fully agree that this section was very reduced. Therefore, more precise information has been incorporated. In order to deeply explain the process followed herein for the contextualization of the findings as well as the generation of the main table, the section materials and methods has been rewritten following PRISMA guidelines to include information regarding the exact search terms and databases queries, as well as detail the inclusion and exclusion criteria and filters applied.
Additionally, a new section has also been generated (point 3) to deal with the limitation of the results regarding the different proteomic techniques employed in the articles included in the review. Detailed information in this regard has also been included in Supplemental Table 1.
Minor point: “Authors correctly mention that CSF is a treasure chest for biomarker discovery in neurodegenerative disease. One interesting implication is whether the invasiveness of this testing can be reduced by considering proteins whose plasma abundance is in high concordance with CSF levels and for which sensitive assays have been developed recently to adequately quantify their presence (e.g., proliferating NFL assays) even at low abundance levels”
We completely agree with the relevance of this topic and we have also explored literature on blood/plasma proteomics in order to propose consistent biomarkers that could be potentially found also in CSF. Unfortunately, there is not much correlation between AD biomarkers found in both biofluids. Indeed, after comparison of our database with the information included in two recent reviews of plasma proteomics in AD samples, we only found three proteins that followed CSF patterns: APOA1 (1)), NFL and CH3L1(2). This observation together with the fact that several proteins behave oppositely in CSF and plasma and the lack of consistent results in plasma made us to leave out these results since we believe that they could difficult the interpretation of our data.
[1] S.H. Rehiman, S.M. Lim, C.F. Neoh, A.B.A. Majeed, A.V. Chin, M.P. Tan, S.B. Kamaruzzaman, K. Ramasamy, Proteomics as a reliable approach for discovery of blood-based Alzheimer's disease biomarkers: A systematic review and meta-analysis, Ageing Res Rev 60 (2020) 101066.
[2] T. Lashley, J.M. Schott, P. Weston, C.E. Murray, H. Wellington, A. Keshavan, S.C. Foti, M. Foiani, J. Toombs, J.D. Rohrer, A. Heslegrave, H. Zetterberg, Molecular biomarkers of Alzheimer's disease: progress and prospects, Dis Model Mech 11(5) (2018).
Round 2
Reviewer 1 Report
I am satisfied with the changes the authors made to the manuscript.
Reviewer 3 Report
The authors have addressed my concerns, and I am happy to recommend this manuscript for publication.